# A Dual-Reporter Platform for Screening Tumor-Targeted Extracellular Vesicles

**DOI:** 10.3390/pharmaceutics14030475

**Published:** 2022-02-22

**Authors:** Masamitsu Kanada, Lauren Linenfelser, Elyssa Cox, Assaf A. Gilad

**Affiliations:** 1Institute for Quantitative Health Science and Engineering (IQ), Michigan State University, East Lansing, MI 48824, USA; linenfe2@msu.edu (L.L.); coxelyss@msu.edu (E.C.); 2Department of Pharmacology & Toxicology, Michigan State University, East Lansing, MI 48824, USA; 3Department of Chemical Engineering & Materials Science, Michigan State University, East Lansing, MI 48824, USA; 4Department of Radiology, Michigan State University, East Lansing, MI 48824, USA

**Keywords:** extracellular vesicles, bioluminescence resonance energy transfer, tumor-homing peptide, lipid nanoprobes, fluorescence microscopy, breast cancer

## Abstract

Extracellular vesicle (EV)-mediated transfer of biomolecules plays an essential role in intercellular communication and may improve targeted drug delivery. In the past decade, various approaches to EV surface modification for targeting specific cells or tissues have been proposed, including genetic engineering of parental cells or postproduction EV engineering. However, due to technical limitations, targeting moieties of engineered EVs have not been thoroughly characterized. Here, we report the bioluminescence resonance energy transfer (BRET) EV reporter, PalmReNL-based dual-reporter platform for characterizing the cellular uptake of tumor-homing peptide (THP)-engineered EVs, targeting PDL1, uPAR, or EGFR proteins expressed in MDA-MB-231 breast cancer cells, simultaneously by bioluminescence measurement and fluorescence microscopy. Bioluminescence analysis of cellular EV uptake revealed the highest binding efficiency of uPAR-targeted EVs, whereas PDL1-targeted EVs showed slower cellular uptake. EVs engineered with two known EGFR-binding peptides via lipid nanoprobes did not increase cellular uptake, indicating that designs of EGFR-binding peptide conjugation to the EV surface are critical for functional EV engineering. Fluorescence analysis of cellular EV uptake allowed us to track individual PalmReNL-EVs bearing THPs in recipient cells. These results demonstrate that the PalmReNL-based EV assay platform can be a foundation for high-throughput screening of tumor-targeted EVs.

## 1. Introduction

Extracellular vesicles (EVs) provide a natural delivery system that transfers various pieces of cellular cargo (protein, mRNA, non-coding RNA, and DNA) to both adjacent and distant cells in the body [1]. EV-based drug delivery has been an active research field in the past decade [2], and previous preclinical studies have demonstrated the superiority of EV-based drug delivery over delivery using synthetic nanocarriers, such as liposomes [3,4].

EVs are heterogeneous populations of nanoscale membrane-enclosed vesicles released virtually from all cell types. At least two main EV classes are released from mammalian cells in the physiological condition: exosomes (~30–120 nm in diameter), formed by inward budding of the endosomal membrane, producing multivesicular bodies (MVBs) that fuse with the plasma membrane [5]; microvesicles (50–1000 nm in diameter), formed by outward budding at the plasma membrane [1]. EVs offer advantages for surface engineering while possessing inherent immune evasion and tissue-penetrating characteristics [3,6,7]. Several approaches have achieved targeted delivery of engineered EVs using various targeting moieties, including antibodies, nanobodies, and peptides [8]. As shown in pioneering studies, EVs have been modified with tissue-targeting peptides by genetically engineering the parental cells—for example, Lamp2b was fused to neuron-specific rabies viral glycoprotein (RVG) peptides [9] or angiogenic vasculature-specific iRGD peptides [10]. Similarly, EGFR-targeting GE11 peptides were genetically fused to the transmembrane domain of the platelet-derived growth factor receptor [11]. As alternative approaches for functionalizing the EV surface with peptides, both covalent and noncovalent postproduction modifications have been proposed—for example, the cyclo(Arg-Gly-Asp-d-Tyr-Lys) peptides were coupled to the EV surface through bio-orthogonal copper-free azide alkyne cycloaddition for targeting ischemic reactive cerebral vascular endothelial cells [12] and brain tumors [13]. Moreover, a recent study demonstrated the covalent conjugation of EVs with high copy numbers of GE11 peptides using protein ligases for targeting EGFR-positive lung cancer cells both in vitro and in vivo [14]. Various approaches for noncovalent postproduction modification of EVs have succeeded. For example, EVs were engineered with stearylated artificial leucine zipper K4 peptides or arginine-rich cell-penetrating peptides (CPPs) to induce active macropinocytosis in vitro [15,16]. More recently, cardiac homing peptide (CHP)-coupled lipid nanoprobes were inserted into an EV membrane for targeting the infarcted heart in murine models [17,18].

Despite the recent advances in the field, a sensitive and quantitative assay platform for screening engineered EVs is still lacking. Lipophilic fluorescent dyes such as PKH, DiR, and DiI are often used to assess the cellular uptake of EVs due to their simple labeling method [19]. While their bright fluorescence may be advantageous for visualizing individual EVs, quantitative assessment of cellular uptake of labeled EVs by bulk measurement (e.g., plate readers) is still a challenge due to the background autofluorescence in culture media. Alternatively, bioluminescent reporters (*Gaussia* luciferase [Gluc] [20,21] and NanoLuc [22,23]) have been developed for assessing in vivo biodistribution, EV release, and cellular EV uptake. Moreover, a recent study demonstrated a bioluminescence resonance energy transfer (BRET)-based EV reporter, PalmGRET [24], for multiresolution imaging of reporter EVs. PalmGRET consists of an eGFP-NanoLuc fusion reporter protein (GpNluc [25]) and a palmitoylation signal sequence [26]. Notably, because of the acid-quenching mechanism [27], the most widely used eGFP is acidic pH-sensitive (p*K*a 6.0). Since recent studies revealed that endocytosed EVs were rapidly translocated into endosomal compartments [28,29], acidic pH-sensitive eGFP (p*K*a 6.0) is not an ideal reporter for tracking cellular EV uptake. Therefore, in this study, we used a novel BRET EV reporter, PalmReNL [30], consisting of a tdTomato-NanoLuc fusion protein [31] (p*K*a for tdTomato: 4.7) and a palmitoylation signal sequence [26], to characterize cellular EV uptake as a dual-reporter platform. We developed tumor-targeted EVs using several tumor-homing peptides (THPs) [32] coupled to lipid nanoprobes for noncovalent postproduction EV engineering, and assessed their increased capability of cancer cell targeting by bioluminescence measurement and fluorescence microscopy in vitro. The PalmReNL-based assay system provides a foundation for high-throughput screening of engineered EVs for tumor-targeted drug delivery.

## 2. Materials and Methods

### 2.1. Plasmid DNA Constructs

For the EV reporter, PalmReNL, a palmitoylation sequence (MLCCMRRTKQ) [26,33] was genetically fused to Red enhanced Nano-lantern (ReNL [31]; Addgene plasmid (Watertown, MA, USA) #89536, gift from Takeharu Nagai) by PCR and subcloned into the Sleeping Beauty transposon vector [34] as we previously reported [30]. HEK293FT cells were cotransfected with a plasmid encoding PalmReNL and pCMV(CAT)T7-SB100 (Addgene #34879, gift from Zsuzsanna Izsvak).

### 2.2. Cell Culture

MDA-MB-231 cells (ATCC) and HEK293FT cells (R700-07, Invitrogen (Waltham, MA, USA)) were cultured in Dulbecco’s modified eagle medium supplemented with GlutaMax (Gibco), 10% (vol/vol) FBS and 1% penicillin/streptomycin and incubated at 37 °C in a 5% CO_2_ atmosphere. HEK293FT cells stably expressing PalmReNL were selected in the presence of puromycin (2 µg/mL) to establish cells constitutively expressing PalmReNL. We further enriched cells highly expressing PalmReNL by limiting dilution.

### 2.3. EV Isolation

EV-depleted FBS was prepared by 18-h ultracentrifugation at 100,000× g, 4 °C [35]. PalmReNL-HEK293FT cells were seeded at 1.5 × 10^6^ cells per 100 mm cell culture dish and cultured for 2–3 d in 10 mL of EV-depleted media. The conditioned medium from a 100 mm cell culture dish was centrifuged at 600× g for 15 min to remove cells and debris. The supernatant was centrifuged at 2000× g for 20 min at room temperature (RT) to remove apoptotic bodies. The supernatant was filtered through a 0.2 µm PES membrane filter (Nalgene (Rochester, NY, USA), 725–2520) with pressure to remove large vesicles. EVs were collected by a size-based EV isolation method [36,37] using 50 nm nanoporous membrane filters (Whatman (Maidstone, UK), WHA110603) with holders (EMD Millipore (Burlington, MA, USA), SX0002500). Holders with 50 nm membrane filters were connected to a vacuum manifold (Qiagen (Hilden, Germany)) and washed with 10 mL of PBS by applying vacuum. The remaining EVs in the supernatant were trapped on the membranes. Then, the enriched EVs were washed with 5 mL of PBS by applying vacuum and 500–1000 µL of samples were carefully collected. EV protein concentrations were measured by the Bradford assay (Thermo Fisher (Waltham, MA, USA)). Bioluminescence and fluorescence signals in the reporter EVs were measured by the bioluminescence (Tecan, Männedorf, Switzerland or BioTek, Winooski, VT, USA) and fluorescence (Molecular Devices, San Jose, CA, USA) plate readers. All EVs were aliquoted and stored at −80 °C.

### 2.4. Nanoparticle Tracking Analysis (NTA)

EVs derived from PalmReNL-HEK293FT cells were analyzed using the ZetaView Multiple Parameter Particle Tracking Analyzer (Particle Metrix, Meerbusch, Germany) following the manufacturer’s instructions. The enriched PalmReNL-EVs were diluted 100- to 1000-fold with PBS for the measurement of particle size and concentration. The instrument was set to the camera sensitivity = 85, Shutter = 250, and frame rate = 30.

### 2.5. EV Membrane Modification

PalmReNL-EVs were engineered with peptides known as THPs (a uPAR-binding peptide: VSNKYFSNIHWGC [38]; a PDL1-binding peptide: YASYHCWCWRDPGRS [39,40]; an EGFR-binding peptide (KKKGG-GE11): KKKGGYHWYGYTPQNVI [41,42]; and an EGFR-binding peptide #2: YHWYGYTPENVI [43]). DOPE-NHS (dioleoylphosphoethanolamine N-hydroxysuccinimide; COATSOME FE-8181SU5, NOF America, White Plains, NY) was coupled to synthesized THPs (GenScript, Piscataway, NJ, USA) for self-insertion of the THP-lipid nanoprobes into EV membranes as previously reported with slight modifications [18]. The THP and DOPE-NHS were dissolved in DMSO at 1 mg/mL and 0.25 mg/mL, respectively. The DOPE-NHS and THP were combined with a 100-fold molar excess of THPs in the presence of an equal vol of HEPES buffer (25 mM, pH 7.5–1.3 nmol) DOPE-NHS plus 130 nmol THPs with 12.5 mM HEPES—and allowed to react at RT for 1 h to form the DOPE-peptides. Then, 10% (vol/vol) Tris buffer (100 mM, pH 7.5) was added to stop the reaction. A negative control, DOPE-lipid nanoprobe only, was prepared in the same condition without THPs. PalmReNL-EVs (1.5 × 10^9^) were then added and incubated with DOPE-THPs or DOPE only at 37 °C for 30 min. The engineered EVs were concentrated using 100 kDa ultrafiltration spin filter (UFC810024, MilliporeSigma (Burlington, MA, USA)) and washed with PBS twice by centrifugation. The engineered EVs (250 µL) were stored at −80 °C.

### 2.6. Liquid Chromatography/Mass Spectrometry (LC/MS)

DOPE-THPs were analyzed by LC/MS using a Thermo Q-Exactive interfaced with a Thermo Vanquish UHPLC (Thermo Scientific, Waltham, MA). Peptides in a solution of 25% DMSO, 10 mM Tris and 25 mM HEPES in water were diluted 1:2 with isopropanol/acetonitrile/water (2:2:1 *v*/*v*/*v*). Then, 10 uL of diluted sample was injected onto a Waters Acquity BEH-C18 UPLC column (2.1 × 100 mm) and peptides were separated with the following gradient: initial conditions were 80% mobile phase A (60% acetonitrile/40% water containing 10 mM ammonium formate and 0.1% formic acid) and 20% mobile phase B (90% isopropanol/10% acetonitrile containing 10 mM ammonium formate and 0.1% formic acid), ramp to 43% B at 2 min, ramp to 54% B at 12 min, ramp to 70% B at 12.1 min, ramp to 99% B at 18 min, return to 20% B at 18.1 min and hold at 20% B until 20 min. The column was held at 55 °C at the flow rate of 0.4 mL/min. Compounds were ionized by electrospray operating in positive ion mode with a capillary voltage of 3.5 kV, transfer capillary temp 262.5 °C, sheath gas set at 50, aux gas set at 12.5, probe heater set to 425 °C, and S-lens RF level at 50. Data were acquired using a full scan method with a m/z range of 500–2000, AGC target of 3 × 10^6^, maximum inject time of 200 ms and resolution set at 70,000. Data were analyzed using QualBrowser in the Thermo Xcalibur software (Version 4.1, Thermo Scientific, Waltham, MA).

### 2.7. Fluorescence Microscopy and Bioluminescence Measurements

The uptake of PalmReNL-EVs by MDA-MB-231 cells was analyzed by fluorescence microscopy and bioluminescence measurement. The cells were plated in 96-well black/clear bottom cell culture plates at a concentration of 45,000 cells/well and cultured for 24 h. The culture medium was replaced with EV-depleted medium and the THP-engineered PalmReNL-EVs were added at a concentration of 6.0 × 10^8^ or 6.0 × 10^9^ EVs/mL. At the end of the incubation period, the cells were washed twice with PBS and the uptake of the reporter EVs was analyzed by measuring bioluminescence after adding furimazine (Fz; 25 μM) using a Spark Multimode Microplate Reader (Tecan, Männedorf, Switzerland) or a Cytation 5 Cell Imaging Multimode Reader (BioTek, Winooski, VT, USA). Phase contrast and fluorescence images of PalmReNL-HEK293FT cells, PalmReNL-EVs, or MDA-MB-231 cells were taken using All-in-one Fluorescence Microscope BZ-X700 (KEYENCE (Osaka, Japan)). For assessing cellular uptake of PalmReNL-EVs, MDA-MB-231 cells in a glass bottom chamber slide (µ-Side 8 Well Glass Bottom, ibidi) were incubated with PalmReNL-EVs with or without THPs (6.0 × 10^9^ EVs/mL). The cells were washed twice with PBS and stained with 10 µg/mL Hoechst 33342 (H3570, Life Technologies (Carlsbad, CA, USA)) before fluorescence microscopy was performed using a DeltaVision Microscope (GE Healthcare Life Sciences, Piscataway, NJ, USA). The fluorescence filter set DAPI/TRITC was used to acquire images and process z-stacks for deconvolution. All images were further analyzed using ImageJ software (Version 1.53a, NIH, Bethesda, MD, USA).

### 2.8. Western Blotting

Whole cell lysates and EVs (4 × 10^8^ EVs) were derived from PalmReNL-HEK293FT cells and mixed with 4× sample buffer (Bio-Rad (Hercules, CA, USA)) with β-mercaptoethanol for detecting Alix, CD9, TSG101, and tdTomato proteins. Whole cell lysates derived from MDA-MB-231 cells were mixed with 2× sample buffer without β-mercaptoethanol (non-reducing conditions) for detecting PDL1, uPAR, EGFR, and GAPDH. Proteins were separated on a 4–20% Mini-PROTEAN TGX gel (Bio-Rad) and transferred to a polyvinylidene difluoride membrane (Millipore, IPFL00010). After blocking with 5% non-fat dry milk at RT for 1 h, membranes were probed with primary antibodies overnight at 4 °C at dilutions recommended by the suppliers as follows: anti-Alix (Proteintech (Rosemont, IL, USA), 12422-1-AP), anti-TSG101 (Proteintech, 14497-1-AP), anti-CD9 (Proteintech, 60232-1-lg), anti-RFP (Rockland Immunochemicals (Pottstown, PA, USA), 600-401-379), anti-PDL1 (Proteintech, 66248-1-lg), anti-uPAR (Proteintech, 10286-1-AP), anti-EGFR (Proteintech, 18986-1-AP), or anti-GAPDH (Santa Cruz (Santa Cruz, CA, USA), G-9), followed by incubation with horseradish peroxidase (HRP)-conjugated secondary antibodies at RT for 1 h. The membranes were visualized with ECL select Western Blotting Detection Reagent (GE Healthcare (Chicago, IL, USA), RPN2235) on ChemiDoc MP Imaging System (Bio-Rad).

### 2.9. Transmission Electron Microscopy

The samples were prepared as previously reported with slight modifications [37,44]. Isolated EVs (PalmReNL-EVs with or without THPs: 6.0 × 10^9^ EVs/mL) were fixed in 1% paraformaldehyde at RT for 30 min. For negative staining of EVs, 5 µL of the sample solution was placed on a carbon-coated EM grid, and EVs were immobilized for 1 min. Next, the samples were washed by placing each one face down on top of a 100 µL droplet of water (6×). The samples were negative stained with 1% uranyl acetate. Excess uranyl acetate was removed by contacting the grid edge with filter paper and the grid was air-dried. Samples were observed using a JEOL 1400 Flash Transmission Electron Microscope (JEOL USA, Peabody, MA, USA) equipped with an integrated Matataki Flash sCMOS bottom-mounted camera. The 1400 Flash was operated at 100 kV.

### 2.10. Statistical Analyses

All data are presented as mean ± SD. Statistical analysis was performed using Prism 9 (GraphPad Software (San Diego, CA, USA)). A two-tailed Student’s *t*-test was used for comparisons between two groups. A value of *p* < 0.05 was considered significant and is indicated on all figures with asterisks (*p* < 0.05 is represented by *, *p* < 0.01 by **, *p* < 0.001 by ***, and *p* < 0.0001 by ****).

## 3. Results

### 3.1. Characterization of Dual-Reporter EVs Derived from HEK293FT Cells Stably Expressing PalmReNL

We have previously utilized PalmReNL [30] as a dual-reporter for bioluminescence and fluorescence tracking of EVs in vitro and in vivo. In the current study, we generated HEK293FT cells stably expressing PalmReNL and selected the EV donor cells that maintained a high level of PalmReNL expression (Figure 1A,B). We next isolated PalmReNL-EVs from the conditioned medium using 50 nm membrane filters as we previously reported [36,37] and characterized them by nanoparticle tracking analysis (NTA). The concentration of PalmReNL-EVs was 1.0 × 10^10^ EVs/mL and their mean diameter was 117.6 nm (Figure 1C). The protein concentration of 1.0 × 10^10^ EVs/mL PalmReNL-EVs was 557 µg/mL. Western blotting analysis of exosome markers, including Alix, CD9, and TSG101, in the whole-cell lysate and isolated PalmReNL-EVs, confirmed successful EV enrichment in our experimental setting (Figure 1D and Appendix A). In addition, tdTomato proteins were detected in both the whole-cell lysate and PalmReNL-EVs.

Next, we measured bioluminescence signals in different concentrations of PalmReNL-EVs using furimazine (25 µM) as the substrate (Figure 1E). The bioluminescence signals were proportional to the concentrations of PalmReNL-EVs, which allows us to assess relative EV amounts by measuring bioluminescence. As a comparison, we measured fluorescence signals in PalmReNL-EVs using a plate reader by exciting tdTomato as ReNL is a fusion protein of tdTomato and NanoLuc [31]. The lowest concentration of detectable PalmReNL-EVs was 3.0 × 10^9^ EVs/mL, and fluorescence signals showed poor linearity (Figure 1F). In addition, individual PalmReNL-EVs were visualized by fluorescence microscopy (Figure 1G), while we cannot distinguish single reporter EVs and their aggregates in this resolution. These results demonstrated that PalmReNL-EVs allow for rapid quantification of isolated reporter EVs as well as cellular EV uptake in vitro. Moreover, PalmReNL can be used as a fluorescent EV reporter for standard microscopy and flow cytometry as well. We previously demonstrated that the labeling efficiency of PalmReNL was 1.2% for small EVs as determined by flow cytometry [30]. However, individual EVs carry probe molecules not detectable by flow cytometry due to the background autofluorescence. Hence, their labeling efficiency may be higher as assessed by bioluminescence measurement [30].

### 3.2. Engineering the Surface of PalmReNL-EVs with Tumor-Homing Peptides (THPs)

Engineering the surface of EVs with tumor-targeting ligands has recently shown significant success by improving the tumor cell-binding capability of EVs [10]. Recent studies have developed a lipid nanoprobe system for rapidly isolating EVs [45] and directing therapeutic EVs to the infarcted heart in animal models [17,18]. In this study, we engineered PalmReNL-EVs with well-characterized three THPs, which bind Urokinase-type plasminogen activator receptor (uPAR; VSNKYFSNIHWGC [38]), Programmed death ligand 1 (PDL1; YASYHCWCWRDPGRS [39,40]), or Epidermal growth factor receptor (EGFR; KKKGGYHWYGYTPQNVI [41,42]) (Figure 2A). LC/MS analysis of the DOPE-conjugated THPs confirmed the successful reaction of DOPE-NHS and THPs in our experimental conditions. Notably, we also found various disulfide linkages of cysteine residues in uPAR- and PDL1-binding peptides, as well as multiple DOPE-conjugation in EGFR-binding peptides likely via lysine residues (Appendix A). First, we characterized the peptide-engineered PalmReNL-EVs by transmission electron microscopy (TEM) (Figure 2B). A heterogeneous mixture of PalmReNL-EVs showed artifactual cup-shaped morphology [46]. There was no noticeable morphological change in the EVs engineered with the uPAR-binding peptide-coupled lipid nanoprobes compared to PalmReNL-EVs without membrane modification (Figure 2B). We further characterized the peptide-engineered PalmReNL-EVs by NTA (Appendix A). Their sizes and concentrations were similar to control PalmReNL-EVs except for the reporter EVs with PDL1-binging peptides, which slightly increased the larger EV population with a higher EV concentration, possibly due to peptide secondary structures or aggregation.

The peptide-engineered PalmReNL-EVs were tested on MDA-MB-231 breast cancer cells in vitro. MDA-MB-231 cells display a mesenchymal-like phenotype representing a good model for metastatic breast cancer [47] (Figure 2C). Previous studies reported that three target molecules, PDL1 [40], uPAR [38], and EGFR [43], were highly expressed in human breast cancer cells. Before carrying out binding assays, we confirmed the high protein expressions of PDL1, uPAR, and EGFR in MDA-MB-231 cells by Western blotting (Figure 2D and Appendix A). All the target membrane proteins were highly expressed in MDA-MB-231 cells, indicating that they are ideal targets for characterizing tumor-targeted EVs.

### 3.3. Bioluminescence Analysis of Cellular Uptake of THP-Engineered PalmReNL-EVs by MDA-MB-231 Cells

Cellular uptake of THP-engineered PalmReNL-EVs by MDA-MB-231 cells was assessed by measuring bioluminescence signals. Recipient cells were cultured with two different concentrations of PalmReNL-EVs (6.0 × 10^8^ or 6.0 × 10^9^ EVs/mL) with or without peptides targeting uPAR, PDL1, or EGFR for 4 h before measuring bioluminescence (Figure 3A). A higher concentration (6.0 × 10^9^ EVs/mL) of PalmReNL-EVs bearing PDL1- or uPAR-binding peptides increased their uptake by MDA-MB-231 cells by 1.9- and 22.5-fold relative to PalmReNL-EVs engineered with DOPE-lipid nanoprobe only, whereas a lower concentration (6.0 × 10^8^ EVs/mL) increased by 2.3- and 8.1-fold, respectively (Figure 3A). This result demonstrated that a higher EV concentration was better for testing PalmReNL-EVs bearing uPAR-binding peptides with higher cellular uptake efficiency. On the other hand, the two EV concentrations tested did not affect the relative cellular uptake efficiency of PalmReNL-EVs bearing PDL1-binding peptides compared to PalmReNL-EVs with DOPE-lipid nanoprobe only.

Of note, PalmReNL-EVs engineered with the EGFR-binding peptide GE11 did not significantly increase their cellular uptake in our experimental conditions. We hypothesized that the affinity of GE11 peptides might be reduced by inserting into the EV membrane. To test this possibility, we assessed PalmReNL-EVs engineered with another EGFR-binding peptide sequence #2 (YHWYGYTPENVI) that previously showed the improved EGFR-binding efficiency as fluorescently labeled peptides [43]. However, PalmReNL-EVs engineered with the EGFR-binding peptide #2 did not increase their uptake by MDA-MB-231 cells (Figure 3B). This result suggests that engineering EVs with these EGFR-binding peptides via DOPE-lipid nanoprobes may disturb their binding with EGFR proteins on the surface of cancer cells or may not have sufficient stability of the EGFR-binding peptides on the EV surface.

### 3.4. Bioluminescence Analysis of Time-Dependent Cellular Uptake of THP-Engineered PalmReNL-EVs

We next evaluated cellular uptake of THP-engineered PalmReNL-EVs by MDA-MB-231 at various time points by measuring bioluminescence signals. The recipient cells were cultured with PalmReNL-EVs (6.0 × 10^9^ EVs/mL) bearing THPs or DOPE-lipid nanoprobes only for 4, 8, 16, and 24 h before bioluminescence analysis. Interestingly, MDA-MB-231 cells cultured with PalmReNL-EVs bearing uPAR-binding peptides showed a rapid decrease of bioluminescence signals. Their signals were 22.1-, 11.3-, 5.9-, and 7.57-fold higher than the cells cultured with control PalmReNL-EVs (DOPE-lipid nanoprobes only) at 4, 8, 16, and 24 h, respectively (Figure 4A,B). On the other hand, PalmReNL-EVs bearing PDL1-binding peptides slowly increased bioluminescence signals over time—1.34-, 2.2-, 3.0-, and 1.89-fold higher bioluminescence signals at 4, 8, 16, and 24 h, respectively (Figure 4C), relative to PelmReNL-EVs with DOPE-lipid nanoprobes only (Figure 4A). PalmReNL-EVs bearing EGFR-binding peptides (GE11) did not show a significant increase in their uptake by MDA-MB-231 cells even after longer culture periods (Figure 4D).

A previous study demonstrated the loss of NanoLuc enzyme activity with endosomal translocation [48]. We also found that PalmReNL-EVs decreased their activity at pH below 6.0 either with or without detergent treatment [30]. Thus, the signal loss of PalmReNL-EVs bearing uPAR-binding peptides with higher binding efficiency at long culture periods may be due to the acidic pH sensitivity of PalmReNL in endosomal compartments. However, PalmReNL-EVs bearing PDL1-binding peptides slowly increased the bioluminescence signal in the recipient cells over the long culture periods, possibly due to continuous cellular uptake of PalmReNL-EVs bearing PDL1-binding peptides with lower binding efficiency. In addition, PalmReNL-EVs engineered with uPAR-binding and PDL1-binding peptides could have different fates in endosomal trafficking.

### 3.5. Fluorescence Analysis of Cellular Uptake of THP-Engineered PalmReNL-EVs

We further characterized cellular uptake of PalmReNL-EVs bearing uPAR-, PDL1-, or EGFR-binding peptides by assessing tdTomato fluorescence (Figure 5). Importantly, in PalmReNL-EVs, NanoLuc enzyme activity and tdTomato fluorescence property exhibit different acidic pH sensitivities [49,50]. While most fluorescent proteins are sensitive to acidic cellular environments, such as late endosomes (pH 5.5–6.0) and lysosomes (pH 4.5–5.5), tdTomato fluorescence (p*K*a 4.7) is less sensitive to acidic pH than eGFP (p*K*a 6.0) [50]. In addition, we previously revealed that PalmReNL-EVs retained better fluorescence signals as assessed by flow cytometry in the recipient cells at the 24 h culture period compared to bioluminescence signals [30].

We cultured MDA-MB-231 cells in EV-depleted media for 8, 16, and 24 h with PalmReNL-EVs (6.0 × 10^9^ EVs/mL) bearing THPs or DOPE-lipid nanoprobes only as a negative control and examined individual reporter EVs in the recipient cells by fluorescence microscopy. Punctate fluorescence signals were observed in the recipient cells cultured with PalmReNL-EVs bearing uPAR- and PDL1-binding peptides for 8 h (Figure 5C,D), compared to PalmReNL-EVs with DOPE-lipid nanoprobes only or EGFR-binding GE11 peptides (Figure 5A,B). Notably, we also detected similar punctate fluorescence signals in the recipient cells cultured with PalmReNL-EVs bearing uPAR- and PDL1-binding peptides both at the 16 h and 24 h culture periods (Figure 5G,H,K,L), whereas MDA-MB-231 cells incubated with PalmReNL-EVs bearing DOPE-lipid nanoprobes only or EGFR-binding peptides showed no significant tdTomato fluorescence signals even after longer culture periods (Figure 5E–J).

## 4. Discussion

We developed a PalmReNL-based dual-reporter platform for characterizing tumor-targeted EVs by bioluminescence measurement and fluorescence microscopy. Bioluminescence measurement enables highly sensitive and quantitative reporter EV detection with negligible background signals. In addition, fluorescence analysis of PalmReNL-EVs enables us to track individual engineered EVs in recipient cancer cells. Moreover, since tdTomato is less sensitive to acidic pH relative to eGFP, PlamReNL may be a superior EV reporter for tracking individual reporter EVs in acidic endosomal compartments by standard fluorescence microscopy or flow cytometry.

Recent studies have demonstrated infarct-directed cardiosphere-derived EVs (CDC-EVs), using a CHP [51], to increase the delivery efficacy and decrease the effective dose of intravenously administered CDC-EVs to reduce undesirable off-target effects [17,18]. The CHP-engineered CDC-EVs increased cellular uptake by cardiomyocytes in vitro, and the engineered EVs improved functional recovery in an ischemia/reperfusion rat model by reducing fibrosis, inducing cardiomyocyte proliferation, and promoting angiogenesis [18]. In the current study, we used the same lipid nanoprobe technique for postproduction engineering of EVs with THPs targeting PDL1, uPAR, and EGFR expressed in human breast cancer cells. The same uPAR-binding peptide sequence was recently used for tumor-targeted microRNA delivery by EVs in mouse models (REF). The current study is the first demonstration of EV engineering with the PDL1-binding peptides to the best of our knowledge. Interestingly, we found that EVs engineered with these THPs showed distinct binding characteristics against MDA-MB-231 cells in vitro—PalmReNL-EVs engineered with uPAR-binding peptides showed faster and more efficient cellular uptake, whereas reporter EVs bearing PDL1-binding peptides showed slower and less efficient cellular uptake. While we tested two different EGFR-binding peptide sequences coupled to the DOPE-lipid nanoprobe, PalmReNL-EVs bearing either EGFR-binding peptides did not increase their uptake by MDA-MB-231 cells. Previous studies have demonstrated EV-based EGFR targeting by genetically fusing the transmembrane domain of platelet-derived growth factor receptor to GE11 peptides [11] or by the covalent postproduction modification of EVs with GE11 peptides using protein ligases [14]. Therefore, we concluded that designs of targeting peptide conjugation on the EV surface may be essential for creating functional EGFR-targeted EVs.

We previously revealed that bioluminescence signals in PalmReNL-EVs rapidly decreased after cellular uptake due to their sensitivity to acidic cellular compartments [30]. This unique aspect of PalmReNL allows us to distinguish between EV binding to the cell surface and their uptake by recipient cells, which is challenging to assess using fluorescent EV reporters in a high-throughput setting. In the current study, PalmReNL-EVs engineered with uPAR-binding peptides showed the highest bioluminescence signals at the earlier time point; however, their bioluminescence signals rapidly decreased over the longer culture periods (Figure 4B vs. Figure 4A). This result suggests that uPAR-targeted reporter EVs are rapidly taken up by MDA-MB-231 cells and traffic into the endosome pathway as recently reported on HT1080 fibrosarcoma cell-derived EVs [28]. On the other hand, while PDL1-targeted PalmReNL-EVs showed less efficient cellular EV uptake at the earlier time point, their bioluminescence signals slowly increased even after the longer culture periods (see 8 and 16 h in Figure 4C vs. Figure 4A), suggesting that PDL1-targeted EVs may slowly bind to the cell surface. In addition, EV engineering with PDL1-binding peptides could affect their endosomal trafficking pathways. Since ReNL consists of tdTomato and NanoLuc, we further followed individual THP-engineered PalmReNL-EVs in the recipient MDA-MB-231 cells by fluorescence microscopy. Importantly, we detected consistent punctate fluorescence signals of THP-engineered PalmReNL in the recipient cells during the culture period we tested (Figure 5), possibly since tdTomato is less sensitive to acidic endosomal compartments. While quantitatively analyzing cellular uptake of uPAR-targeted and PDL1-targeted PalmReNL-EVs based on their fluorescence signals is challenging, it allows for long-term tracking of individual tumor-targeted reporter EVs in recipient cells.

## 5. Conclusions

We developed a novel approach for screening THP-engineered EVs using PalmReNL as a dual EV reporter for bioluminescence measurement and fluorescence microscopy in vitro. In this study, three THPs targeting PDL1, uPAR, and EGFR were tested for postproduction engineering of EVs via a DOPE-lipid nanoprobe to increase their binding to MDA-MB-231 cells. We demonstrated that PalmReNL provides an excellent platform for characterizing the cellular uptake efficiency of THP-engineered EVs by measuring bioluminescence signals. Our data further indicate that PalmReNL allows for long-term tracking of individual THP-engineered EVs by fluorescence microscopy.

## Figures and Tables

**Figure 1 pharmaceutics-14-00475-f001:**
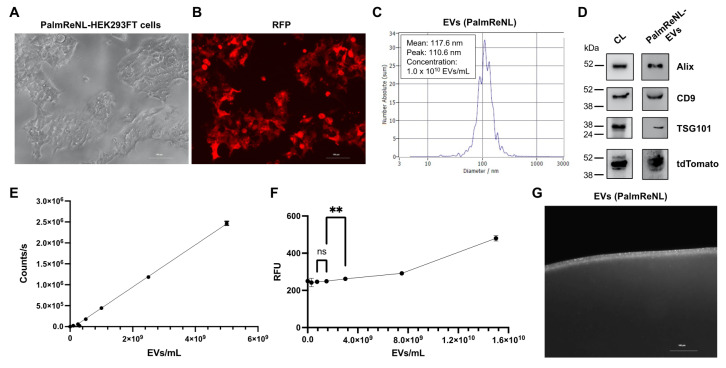
Characterization of dual-reporter PalmReNL-EVs. (**A**) An image of phase contrast in HEK293FT cells stably expressing PalmReNL. (**B**) A fluorescence image of PalmReNL-HEK293FT cells. Scale bar = 100 µm. (**C**) PalmReNL-EVs were analyzed by nanoparticle tracking analysis (NTA). (**D**) Western blot analysis of exosome marker proteins in PalmReNL-HEK293FT cells and -EVs. (**E**) Bioluminescence analysis of PalmReNL-EVs using furimazine. (**F**) Fluorescence signals in PalmReNL-EVs measured by using a plate reader; ns: not significant; **, *p* < 0.01. (**G**) A droplet of buffer containing isolated PalmReNL-EVs visualized by fluorescence microscopy. Scale bar = 10 µm.

**Figure 2 pharmaceutics-14-00475-f002:**
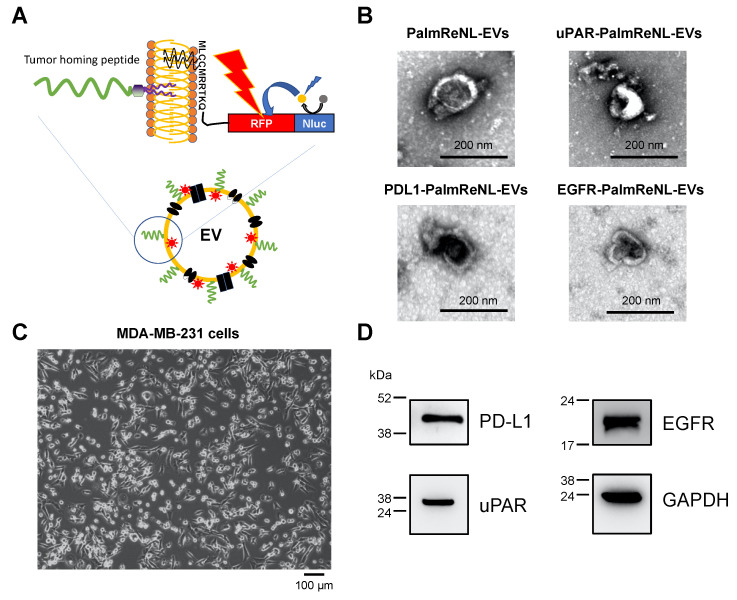
Engineered PalmReNL-EVs with tumor-homing peptides (THPs). (**A**) Schematic diagram of EV membrane labeling with PalmReNL BRET probes and functionalization with THP-coupled lipid nanoprobes. (**B**) Transmission electron microscopy of HEK293FT cell-derived PalmReNL-EVs with or without uPAR-, PDL1-, and EGFR-binding peptides. Scale bar, 200 nm. (**C**) A phase contrast image of MDA-MB-231 cells. Scale bar = 100 µm. (**D**) Western blot analysis of the target membrane protein expression in MDA-MB-231 cells under the non-reducing condition. GAPDH is a loading control.

**Figure 3 pharmaceutics-14-00475-f003:**
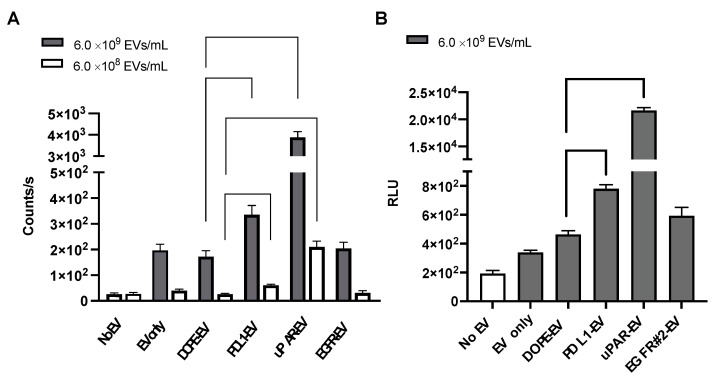
Bioluminescence analysis of cellular uptake of THP-engineered PalmReNL-EVs. (**A**) Bioluminescence analysis of the recipient MDA-MB-231 cells cultured with two different concentrations (6.0 × 10^8^ and 6.0 × 10^9^ EVs/mL) of THP-engineered PalmReNL-EVs using furimazine. (**B**) Bioluminescence analysis of cellular uptake of PalmReNL-EVs (6.0 × 10^9^ EVs/mL) engineered with the EGFR-binding peptide #2, together with PalmReNL-EVs bearing PDL1-, uPAR-binding peptides, or DOPE-lipid nanoprobes only. (**A**,**B**) Error bars, SD (*n* = 5); two-sided Student’s *t*-tests was used for comparison.

**Figure 4 pharmaceutics-14-00475-f004:**
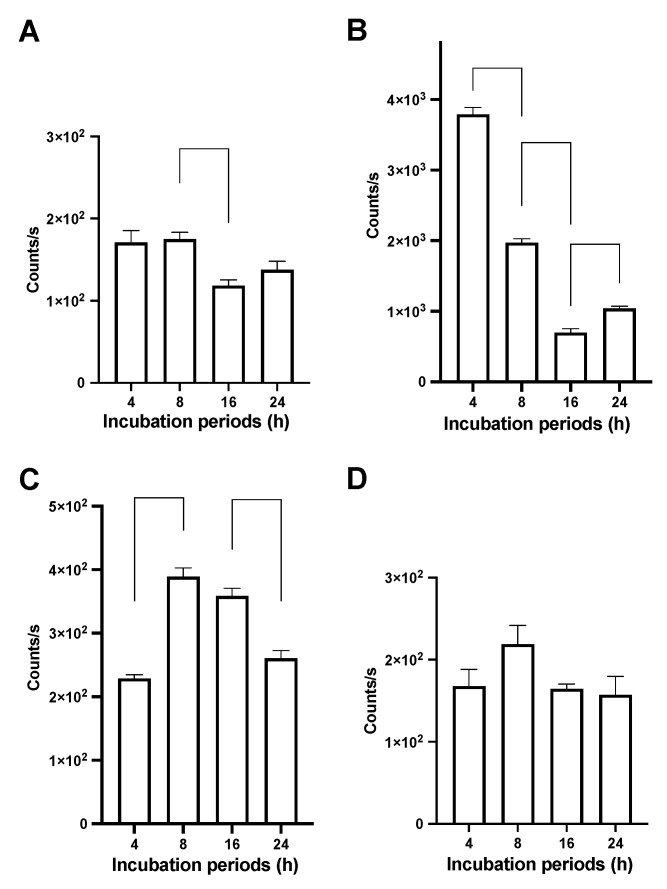
Bioluminescence analysis of time-dependent cellular uptake of THP-engineered PalmReNL-EVs by MDA-MB-231 cells. Bioluminescence analysis of the recipient MDA-MB-231 cells cultured with PalmReNL-EVs bearing DOPE-lipid nanoprobes only (**A**), uPAR-binding (**B**), PDL1-binding (**C**), or EGFR-binding (**D**) at 4, 8, 16, and 24 h post-incubation. (**A**–**D**) Error bars, SD (*n* = 5); two-sided Student’s *t*-test was used for comparison.

**Figure 5 pharmaceutics-14-00475-f005:**
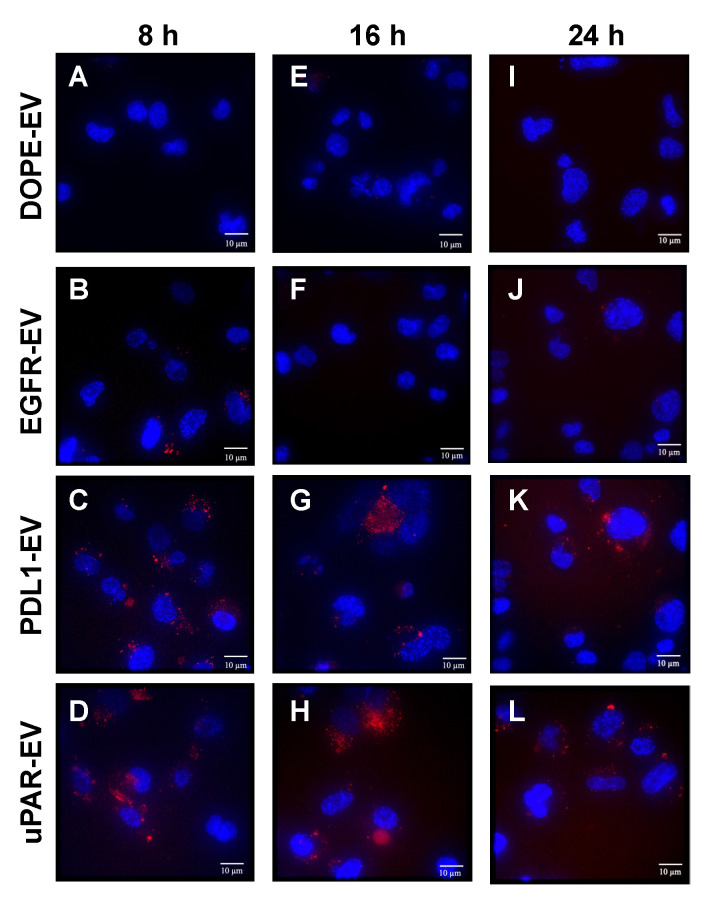
Fluorescence analysis of THP-engineered PalmReNL-EV uptake by MDA-MB-231 cells. MDA-MB-231 cells cultured with PalmReNL-EVs bearing DOPE-lipid nanoprobes only (**A**,**E**,**I**), EGFR- (**B**,**F**,**J**), PDL1- (**C**,**G**,**K**), or uPAR-binding peptides (**D**,**H**,**L**) for 8, 16, and 24 h. Punctate fluorescence signals of tdTomato (red) were merged with nuclei stained with Hoechst 33342 (blue). Scale bar = 10 µm.

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
