# Peer review of "A Dual-Reporter Platform for Screening Tumor-Targeted Extracellular Vesicles"

_pharmaceutics, 2022, doi:10.3390/pharmaceutics14030475_

Round 1

Reviewer 1 Report

This work represents a breakthrough in the characterization of extracellular vesicles, which is important for increasing their clinical use. 

- Lines 133- 134 What are the pH values of TRIS and HEPES buffers?

- Line 198 How did you measure the EV  protein concentrations?

- Line 228 “ There was 228 no noticeable morphological change in the EVs engineered with the uPAR-binding peptide-coupled lipid nanoprobes compared to PalmReNL-EVs without membrane modification” What happened with the other binding peptides?

- Line 253 The different behavior of EV with PDL1-EV and uPAR binding peptide for the two initial concentrations used in the assay could be related to their cell biodistribution?

- Figure 3. The legend of the abscissa in part A of this figure is not legible. It is also not unclear that using the same probes in the EVs the counts are so different between A and B.

Also the figure does not include the * for statistical significance

- Line 284-287 Please rewrite this phase to make it easier to understand:

“Interestingly, MDA-284 MB-231 cells cultured with PalmReNL-EVs engineered with uPAR-binding peptides 285 showed a rapid decrease of bioluminescence signals—22.1-, 11.3-, 5.9-, and 7.57-fold 286 higher bioluminescence signals at 4, 8, 16, and 24 h, respectively (Figure 4B), relative to 287 PalmReNL-EVs with DOPE-lipid nanoprobes only (Figure 4A).”

- Figure 4. It does not include the * for statistical significance

-Line 326 and 331 Where is Figure 5?

Reviewer 2 Report

This article describes the uptake of engineered HEK293 EVs into breast cancer cell line. These EVs are derived from cells that express PalmReNL - a dual-reporter protein for bioluminescence and fluorescence.

PalmReNL is an imaging probe developed by authors of the manuscript in their earlier work. This probe consists of palmitoylation signal peptide (Palm) with red-shifted bioluminescence reporter protein. Palmitoylation should anchor ReNL into membranes. ReNL supplemented with its substrate furimazine produces red-shifted luminescence.  When exciting reporter protein using settings for dtTomato, it can be used also as a fluorescent EV reporter, providing more flexibility for further analysis.

PalmReNL EVs were isolated from HEK293 cell conditioned medium and size distribution and concentration was determined by NTA analysis. EV marker expression, such as CD9, ALIX and TSG101, were analyzed in isolated EVs. Also, bioluminescence and fluorescence were detected.

After characterization of isolated reporter EVs, they were conjugated with three different tumor homing peptides targeting PDL1, uPAR, and EGFR proteins. The uptake of peptide-conjugated and control EVs was analyzed in breast cancer cell line, MDA-MB-231 after incubation two different doses of EVs with cells and measuring luminescence and fluorescence after several timepoints of incubation. Authors confirmed uptake of EVs targeting uPAR and PDL1 and EVs that were conjugated with EGFR-targeting peptides did not show any uptake. When evaluating uptake with bioluminescence signals, then uptake of EVs conjugated to uPAR targeting peptide is reduced to background levels (control EVs) and lower during longer incubation timepoints; the uptake of EVs targeting PDL1 is peaking after 8h of incubation and then reducing almost to background levels. When evaluating uptake of these EVs by fluorescence measurements, then uptake could be seen for uPAR targeting EVs after 24h and for PDL1 even after 16h.

This discrepancy has been explained by the pH sensitivity of bioluminescence producing enzyme and even more, it has been suggested as an alternative reporter system due to this property. However, strong evidence supporting both of these ideas are lacking. In light of these findings, authors should reconsider PalmReNL-based EV assay platform as a foundation for high-throughput screening of tumor-targeted EVs.

Q1. What is the concentration and the % of EVs carrying dual-reporter protein in total isolated EV population?

Q2. What is the lowest concentration of EVs that could be detected with bioluminescence and fluorescence?

Comment – in line 206 and 207: individual PalmReNL-EVs were visualized by exciting tdTomato (Figure 1F) as ReNL is a fusion pro-207 tein of tdTomato and NanoLuc. 

It is claimed, that these are individual EVs, however, this resolution should not allow to detect single EVs but rather EV aggregates. To analyze fluorescence of single EVs, dedicated equipment such as fNTA or nanoFCM should be used.

Q3. In line 223-225: In this study, we engineered PalmReNL-EVs with well-characterized three THPs, which bind Urokinase-type plasminogen activator receptor (uPAR), Programmed death ligand 1 (PDL1), or Epidermal growth 225 factor receptor (EGFR).

Q4. Could you please bring out references and sequences that describe these specific peptides out from materials and methods into introduction or results? Secondly, are those peptides used before in conjugation with EVs before and what where results?

Q5. What is the average size and concentration of THP-modified EVs compared to unmodified and DOPE-EVs?

Q6. The uptake of EVs is higher for PDL1 and uPAR targeting peptide conjugated EVs but EGFR targeting peptide conjugated EVs do not show any uptake. Meanwhile, it was confirmed that these cells were expressing target protein. Would it be possible to find out what is the EGFR-targeting peptide conjugation efficiency on the surface of reporter EVs?

Comment: in line 302-303: We also found that PalmReNL-EVs decreased their activity at pH below 6.0 either with or without detergent treatment [29].

Real uptake of therapeutic molecules into cells start with escaping endo-lysosomal pathways - then biologically active molecules can translocate to their intracellular targets. If the activity of reporter system is reduced or diminished during passing this pathway, it indicates its incompatibility for uptake studies. Therefore, following the fluorescence over period of time gives more complete understanding about uptake kinetics of THP-conjugated reporter EVs.

Q7. Would it be possible to quantify intracellular fluorescence activity from cells or cell lysates by fluorimetry after incubation of indicated timepoints? This can serve as a semi-quantitative uptake analysis for modified EVs and could be further compared to luminescence measurements.

Comment: To provide better evidence for THP-EV membrane association kinetics and advancing through endo-lysosomal pathway, microscopy colocalization studies with membrane specific dyes and/or endo-lysosomal markers could give more support for EV PalmReNL reporter system and its cellular uptake characterization.

Comment: In material and methods section – could you please bring out more in detail: NTA settings for measurements, imaging settings for microscopy and specific conditions for uptake studies – if cells were washed or not after incubating with reporter EVs.

Round 2

Reviewer 2 Report

I would like to thank authors for making significant and necessary improvements in manuscript and for answering questions thoroughly.  These improvements had significant effect to the manuscript. As a result, i do not have any further substantive questions or comments.